# Dietary EPA+DHA Mitigate Hepatic Toxicity and Modify the Oxylipin Profile in an Animal Model of Colorectal Cancer Treated with Chemotherapy

**DOI:** 10.3390/cancers14225703

**Published:** 2022-11-21

**Authors:** Md Monirujjaman, Oliver F. Bathe, Vera C. Mazurak

**Affiliations:** 1Division of Human Nutrition, Department of Agricultural Food and Nutritional Science, Li Ka Shing Centre for Health Research Innovation, University of Alberta, Edmonton, AB T6G 2P5, Canada; 2Department of Surgery and Oncology, University of Calgary, Calgary, AB T2N 1N4, Canada

**Keywords:** arachidonic acid, cytokines, CASSH, fish oil, irinotecan, liver, tumor

## Abstract

**Simple Summary:**

Liver toxicity as a result of chemotherapy for metastatic colorectal cancer is common, but it remains poorly understood. The bioactive metabolites of polyunsaturated fatty acids, oxylipins, and pro-inflammatory cytokines play crucial roles in numerous biological processes and pathological conditions. A preclinical model of colorectal cancer treated with chemotherapy was used to understand the alterations of oxylipins and cytokines concurrently with the alterations of the liver pathology and to test whether liver toxicity can be modified by a diet containing eicosapentaenoic acid (EPA) and docosahexaenoic acid (DHA) in the form of fish oil. After chemotherapy, the measures of liver toxicity were evident and were associated with the elevation of specific oxylipins and cytokines. Dietary fish oil influenced oxylipin and cytokine production while mitigating liver toxicity after chemotherapy treatment. Fish oil may present a novel dietary strategy to attenuate liver toxicity during chemotherapy.

**Abstract:**

Irinotecan (CPT-11) and 5-fluorouracil (5-FU) are commonly used to treat metastatic colorectal cancer, but chemotherapy-associated steatosis/steatohepatitis (CASSH) frequently accompanies their use. The objective of this study was to determine effect of CPT-11+5-FU on liver toxicity, liver oxylipins, and cytokines, and to explore whether these alterations could be modified by dietary eicosapentaenoic acid (EPA) and docosahexaenoic acid (DHA) in the form of fish oil (EPA+DHA). Tumor-bearing animals were administered CPT-11+5-FU and maintained on a control diet or a diet containing EPA+DHA (2.3 g/100 g). Livers were collected one week after chemotherapy for the analysis of oxylipins, cytokines, and markers of liver pathology (oxidized glutathione, GSSH; 4-hydroxynonenal, 4-HNE, and type-I collagen fiber). Dietary EPA+DHA prevented the chemotherapy-induced increases in liver GSSH (*p* < 0.011) and 4-HNE (*p* < 0.006). Compared with the tumor-bearing animals, ten oxylipins were altered (three/ten n-6 oxylipins were elevated while seven/ten n-3 oxylipins were reduced) following chemotherapy. Reductions in the n-3 fatty-acid-derived oxylipins that were evident following chemotherapy were restored by dietary EPA+DHA. Liver TNF-α, IL-6 and IL-10 were elevated (*p* < 0.05) following chemotherapy; dietary EPA+DHA reduced IL-6 (*p* = 0.09) and eotaxin (*p* = 0.007) levels. Chemotherapy-induced liver injury results in distinct alterations in oxylipins and cytokines, and dietary EPA+DHA attenuates these pathophysiological effects.

## 1. Introduction

Colorectal cancer (CRC) is the second-most-common cause of cancer death, and it is estimated that it will represent 8–10% of all new cancer cases in 2022 [1,2]. Liver metastasis is common in patients with CRC, as approximately 15–25% of patients present with liver metastases at the time of primary diagnosis and a further 18–25% patients have developed metastases at follow-up [3]. Surgical resection is the most effective treatment, and is curative in 20–25% patients with a median survival of 40 months [4]. Chemotherapy provided prior to surgery reduces tumor size, increases curative resection rates, and sometimes converts an unresectable disease to a resectable disease [4,5]. A common drug combination for colorectal liver metastasis is irinotecan (CPT-11) + 5-fluorouracil (5-FU). While the drug combination has a good response rate, it is associated with steatosis and steatohepatitis [4,5,6,7,8]; related hepatic dysfunction may limit the capability to resect larger portions of the liver [4]. A higher prevalence (20.2% vs. 4.4%, *p* = 0.001) of chemotherapy-associated steatohepatitis was observed in patients with colorectal liver metastasis receiving CPT-11+5-FU-based chemotherapy before liver resection when compared with patients who had no chemotherapy [7]. Although chemotherapy-associated steatosis/steatohepatitis (CASSH) is common, the underlying mechanisms are poorly understood, and are being continually investigated.

Oxylipins are biologically active compounds present in all tissues, including the liver. Oxylipins are primarily formed enzymatically by the oxidative metabolism of polyunsaturated fatty acids (PUFAs) [9,10]. Upon release from membrane phospholipids (PLs) by the action of cytosolic phospholipase A2, free PUFAs are converted to oxylipins by three classes of enzymes (cyclooxygenase, lipoxygenase, and cytochrome P450) via oxygenation reactions to form distinct products, depending on the fatty acid substrate [9,11]. Oxylipins are important regulators of tissue homeostasis, inflammation, and signaling, and are involved in many inflammatory diseases, including arthritis, cancer, cardiovascular disease, and renal disease [9,10,12,13,14]. Oxylipins are key mediators of hepatic inflammation [15,16,17], and therefore might play an important role in the pathogenesis of CASSH.

Depending on the PUFA substrate, oxylipins can be classified as pro- or anti-inflammatory, due to the fact that oxylipins derived from n-3 PUFA are generally anti-inflammatory or less inflammatory, while those derived from n-6 PUFA are generally pro-inflammatory [9,18,19,20]. Pro-inflammatory oxylipins induce the expression of pro-inflammatory cytokines such as interleukin (IL)-1β, IL-6, and tumor necrosis factor (TNF)-α, while anti-inflammatory oxylipins decrease their expression and limit the production of oxylipins that are inflammatory [21,22,23]. Dietary supplementation with specific n-3 PUFAs (i.e., eicosapentaenoic acid, EPA, and docosahexaenoic acid, DHA) increase the cellular membrane’s phospholipid fatty acid content of these fatty acids [24,25,26]. Although the incorporation of EPA+DHA into the membrane alters the substrate available for enzymatic oxidation, the molecular mode of action of PUFAs is still not fully understood; however, it is known that the physiological effects of PUFAs are significantly mediated by the oxylipins produced [9,12,14,16,20,27]. Therefore, to understand the net consequences of any alteration in PUFAs, it is necessary to directly determine what oxylipins are produced.

CASSH has many features similar to non-alcoholic fatty liver disease (NAFLD), and some studies have reported alterations of circulating oxylipin profiles in NAFLD [28,29,30,31,32]. It is also well accepted that inflammatory cytokines play a critical role in contributing to the development and progression of NAFLD [33,34]. However, no study has reported on hepatic oxylipins nor cytokine alterations in CASSH.

It was hypothesized that chemotherapy is associated with both perturbations in oxylipins and increased pro-inflammatory cytokines. Our hypothesis is based on our previous study [35], in which hepatic PUFAs were depleted after chemotherapy treatment in the same animal model. It was also postulated that providing dietary EPA+DHA in the form of fish oil initiated at the same time as chemotherapy was administrated would reduce liver toxicities and mitigate the oxylipin and pro-inflammatory cytokine alterations caused by chemotherapy treatment.

## 2. Materials and Methods

### 2.1. Animal Handling and Experimental Design

A total of 18 female Fischer 344 rats (Charles River, Senneville, QC, Canada) underwent tumor implantation after two weeks of basal diet feeding, as previously described [36]. Tumor volume was estimated daily using the following formula: cm^3^ = 0.5 × L × W × H [37]. When the tumor volume reached 2 cm^3^, one group (*n* = 6) of rats were euthanized, while the remaining rats (*n* = 12) were administrated with intraperitoneal CPT-11 injections (50 mg/kg body weight), while 5-FU (50 mg/kg body weight, intraperitoneal) was injected following the next day of CPT-11 injection. The combination of CPT-11+5-FU represents one complete chemotherapy cycle. The rats were randomized to a diet containing fish oil (designated as the Chemo+Fish Oil group throughout the manuscript; *n* = 6) starting on the day of the CPT-11 injection, or a control semi-purified diet resembling the intake of western diets (designated as the Chemotherapy or Chemo group; *n* = 6) [38]. Tumor-bearing animals not receiving the chemotherapy treatment (*n* = 6) designated as the Tumor group were provided a control diet. Healthy rats (*n* = 6) did not undergo tumor implantation nor receive chemotherapy, consumed only the control diet (designated as the Reference group throughout the manuscript), and were otherwise handled in the same manner as the experimental groups. The experimental design is outlined in Figure 1.

Rats were acclimated for one week prior to the start of the experiment, then fed a North American type of diet based on the AIN-76 basal diet (Harlan Teklad, Indianapolis, IN, USA), while the fish oil diet contained the same proportion of macronutrients as the control diet, differing only in the addition of 2.3 g fish oil/100 g (Ocean Nutrition Canada, Dartmouth, NS, Canada). The food intake and body weight were measured daily following the chemotherapy injections. The rats were euthanized with carbon-dioxide asphyxiation, and their livers were collected, weighed, and immediately snap-frozen in liquid nitrogen and stored at −80 °C for further analysis.

### 2.2. Assessment of Oxidative Stress, Lipid Peroxidation, and Fibrosis Markers from Liver Tissue

To determine the oxidized glutathione (GSSH), the frozen rat livers (~30 mg) were homogenized (1:10 *w*:*v* ratio) in the extraction buffer (supplied with the kit) and determined calorimetrically (Abbkine, Inc, Wuhan, China, Cat#: KTB1610). For the lipid peroxidation marker 4-Hydroxy-2-Nonenal (4-HNE), the frozen livers (~30 mg) were homogenized (1:20 *w*/*v* ratio) using a RIPA buffer containing SDS and determined using a rat ELISA kit (Abcam, Cambridge, UK, Cat#ab238538). For the classic liver fibrosis marker type-I collagen (COL-1), the frozen livers (~30 mg) were homogenized (1:9 *w*/*v* ratio) using PBS (0.01 M, pH = 7.4) and measured using specific rat ELISA kits (Novus Biologicals, Centennial, CO, USA, Cat# NBP2-75823). All procedures were performed according to the manufacturer’s instructions.

### 2.3. Oxylipin Analysis

Oxylipins were analyzed using a previously established method [25,39,40,41,42]. Briefly, whole liver tissues were homogenized in ice-cold Tyrode’s salt solution (pH 7.6) and Triton X-100 was added to achieve a final concentration of 0.01%. Deuterated internal standards (10 µL) were added to 200 µL tissue homogenates to perform solid-phase extraction using Strata-X SPE columns (Phenomenex, Torrance, CA, USA). The samples were loaded onto the columns, washed with 10% methanol and pH 3 water, and dried with hexane, followed by being eluted with 100% methanol. The samples were dried under nitrogen and resuspended in the mobile phase (water/acetonitrile/acetic acid, 70/30/0.02 *v*/*v*/*v*) for HPLC/MS/MS (Shimadzu Nexera XR) coupled with a QTRAP 6500 (Sciex, Concord, ON, Canada) as described in [39,40,41,42], based on methods developed by Deems et al. [43]. Lists of all of the oxylipins screened for, detector response factors, and internal standards are available in [40,42]. The detection and quantification limits were set at 3 and 5 levels above the background, respectively. The quantification of the oxylipins was determined using the stable-isotope-dilution method [44], and the amounts are expressed as pg/mg of tissue.

### 2.4. Phospholipid (PL) Fatty Acid Analysis

The whole rat livers (50 mg) were homogenized with a calcium chloride solution (0.025%) using a sonicator, and the fatty acid analysis was carried out by the previously described method [35,36,38]. A modified Folch method was used for the extraction of the total lipids from the livers [45]. The PL fraction was isolated using thin-layer chromatography and the band was visualized, scraped, and C17:0 (0.05 µg; Supelco, Bellefonte, PA, USA; Sigma Chemical, St. Louis, MO, USA) was added to enable the quantification of the fatty acids in the PL fraction. The esterified fatty acids were determined using gas chromatography (Varian 3600CX Gas Chromatograph) equipped with a flame ionization detector and a BP-20 fused capillary column (SGE Instruments, Melbourne, Australia). The fatty acid contents were calculated as proportions (%) as well as absolute amounts (μg/g) based on commercially available standards containing a known fatty acid composition.

### 2.5. Determination of Cytokines

Frozen whole rat livers (~50 mg) were pulverized in liquid nitrogen using a mortar and pestle and were homogenized (1:10 *w*:*v* ratio) in an extraction buffer (20 mM Tris HCl pH 7.5; 0.5% Tween 20; 150 mM NaCl and protease inhibitors 1:100) with glass beads (0.5 mm diameter; Fast Prep^®^-24, MP Biomedicals, Santa Ana, CA, USA) for 25 s, then were placed on ice. All samples were diluted to the same protein concentration of 1.6 mg/mL. Luminex xMAP technology was performed using the Luminex™ 200 system (Luminex, Austin, TX, USA). Ten cytokines (TNF-α, Eotaxin, IFN-γ, IL-1α, IL-1β, IL-6, IL-10, IL-17A, IL-18, and MCP-1) were simultaneously measured in the samples using the Rat Cytokine Multi Plex Discovery Assay^®^ (Eve Tech, Calgary, AB, Canada), according to the manufacturer’s protocol. The assay sensitivities of these markers ranged from 0.3–30.7 pg/mL for the multiplex.

### 2.6. Statistical Analysis

Data was analyzed by SPSS Statistics for Windows, Version 28.0 (IBM SPSS Statistics for Windows, Version 28.0. IBM Corp, Armonk, NY, USA). The normalities of the data were examined using the Shapiro–Wilk statistic (W > 0.05 for normally distributed data). One-way ANOVA was used to identify significant differences between the groups. The Kruskal–Wallis (nonparametric) test was performed if the data did not follow a normal distribution even if transformed. Post hoc analysis was carried out using the Duncan multiple range test for simple-effect comparisons when interactions were present. All data are presented as mean ± SD. A *p*-value of <0.05 or 0.008 (for oxylipins Bonferroni-corrected *p*-value) was considered a significant difference for the interaction and simple effects.

## 3. Results

### 3.1. General Findings

There were no significant differences in body weights between the treatment groups at the time of termination (152 ± 7.4 g). The relative food intake significantly decreased in the chemotherapy group but returned to baseline by day four, and all groups had similar daily food intakes (10 ± 0.7 g). The average liver weight, as well as the average liver weight as a percent of body weight, was not different between the groups (5.5 ± 0.2 g and 3.7 ± 0.1%, respectively).

### 3.2. Effects of Tumor, Chemotherapy, and Dietary Fish Oil on Liver Toxicity Markers

Liver GSSH and 4-HNE were not affected by the tumor; however, both were significantly (*p* < 0.05) higher following the chemotherapy treatment. For each of these markers, dietary fish oil restored them to values similar to the Reference group (Figure 2A,B). No significant difference (*p* = 0.124) was observed for the fibrosis marker, Col I, between the groups (Figure 2C).

### 3.3. Oxylipin and PL Fatty Acid Distribution

Out of the 160 oxylipins scanned, 76 oxylipins were detected and quantified in the livers. Approximately two-thirds (66%) of the oxylipins were derived from n-6 PUFA, with the majority (72%) being derived from arachidonic acid (AA). Approximately half of the remaining n-3 PUFA-derived oxylipins were formed from DHA, 40% from EPA, and one-tenth (12%) from alpha-linolenic acid (ALA), respectively (Appendix A).

The oxylipin mass did not reflect the PUFA proportions in the PL fractions. For example, in the Reference group, the proportions of linoleic acid (LA), AA, ALA, EPA, and DHA in the PL fraction were 16%, 61%, 0%, 1%, and 19%, respectively, while the proportions of LA, AA, ALA, EPA, and DHA oxylipins were 31%, 54%, 1%, 5%, and 8%, respectively (Figure 3). Similar discrepancies in the distributions of oxylipin masses compared with PUFA mass in PL were also observed in other groups (Figure 3).

### 3.4. Effects of Tumor, Chemotherapy, and Fish Oil on Liver Oxylipins

Compared with the Reference group, seven oxylipins were altered in the livers of the Tumor group, of which six oxylipins (five from AA and one from ALA) were elevated while one EPA-derived oxylipin was reduced (Appendix A).

Ten oxylipins were significantly different between the Tumor and Chemotherapy groups. Three oxylipins derived from AA were higher, while seven oxylipins derived from EPA (two/seven) and DHA (five/seven) were lower following chemotherapy (Appendix A), reflecting the imbalance in n-6/n-3 PUFA metabolites. The total n-3 PUFA-derived oxylipins were also lower (20% reduction by mass) in the liver of the Chemotherapy group when compared with the Tumor group (Figure 4).

In general, n-6 PUFA-derived oxylipins were reduced and n-3 PUFA-derived oxylipins were increased in the livers of the Chemo+Fish Oil group compared with the Chemotherapy group. The heatmap (Figure 5) clearly shows the strong dietary effects of fish oil on liver oxylipins, in which the dietary EPA+DHA reduced the n-6-derived oxylipins while increasing the EPA- and DHA-derived oxylipins (data in Appendix A).

About half (25/51) of the individual n-6 PUFA-derived oxylipins that were detectable in the livers of the Chemotherapy group were reduced in the Chemo+Fish Oil group (Appendix A and visualized in Figure 5). The total, as well as individual, oxylipins derived from LA (5/9), gamma-linolenic acid (1/1), dihomo-γ-linolenic acid (3/4), and AA (16/37) were lower in the Chemo+Fish Oil group (Appendix A). The total n-6 oxylipins were also reduced by half in the livers of the Chemo+Fish Oil group compared with the Chemotherapy group (Figure 4A).

Dietary fish oil containing EPA and DHA increased the total as well individual n-3 oxylipins derived from EPA and DHA (Appendix A and Figure 4B; visualized in Figure 5). All of the EPA and DHA oxylipins were higher in the Chemo+Fish Oil group, except for the DHA-derived 8-HDoHE and the 19,20-DiHDoPE, which remained similar to other groups. Two (out of three) ALA-derived oxylipins were decreased in the Chemo+Fish Oil group compared with the Chemotherapy group (Appendix A). The total n-3 oxylipins were four-fold higher in the livers of the Chemo+Fish Oil group compared with the Chemotherapy group (Figure 4B).

### 3.5. Effects of Tumor, Chemotherapy, and Dietary Fish Oil on Liver Cytokines and Chemokines

Tumors had little effect on liver cytokines with the exception of TNF-α, which was lower in the livers of the Tumor group compared with the Reference group (*p* < 0.05). TNF-α, IL-6, and IL-10 were all significantly higher in the Chemotherapy group compared with Tumor group. Eotaxin, IL-6, and IL-18 levels were lower in the Chemo+Fish Oil group when compared with the Chemotherapy (eotaxin/IL-6) or Tumor (IL-18) groups, respectively (*p* < 0.05, Figure 6A–E and Appendix A). Monocyte chemoattractant protein (MCP)-1 levels were below the detection limit of the assay for all groups.

## 4. Discussion

This work demonstrates a distinct alteration of hepatic oxylipin profiles by a common chemotherapeutic regimen with concurrent increase of markers of liver toxicity and pro-inflammatory cytokines. Each of these alterations are modified in a beneficial manner by dietary EPA+DHA in the form of fish oil. Cytotoxic chemotherapy prolongs the survival rates of patients with advanced and metastatic CRC. However, several cytotoxic agents, including CPT-11 and 5-FU, that are administered routinely have been linked to liver toxicities, such as steatosis and/or steatohepatitis, that impair liver function and regeneration [46,47,48,49]. The exact mechanism(s) of liver toxicity during chemotherapy treatment in CRC patients are still being investigated; however, it is well accepted that inflammation and the production of reactive oxygen species play important roles in liver toxicities [50,51,52,53].

Oxylipins play important roles in regulating inflammation and oxidative stress in diverse tissues, including livers [19,54], by being upstream mediators produced after the cleavage of membrane phospholipids. Although no study has investigated systemic or hepatic oxylipin alterations by chemotherapy treatment, several studies have reported that serum oxylipins can be used as a maker of NAFLD detection or severity [28,30,31,32]. The present study shows that chemotherapy treatment increases several n-6 PUFA oxylipins, in particular those derived from AA, while reducing n-3 PUFA (EPA and DHA)-derived oxylipins. These findings are consistent with previous findings in which AA-derived 12-hydroxy-eicosatetraenoic acid (HETE) was higher, while EPA-derived 11-hydroxy-eicosapentaenoic acid (HEPE) and DHA-derived 13-hydroxy-docosahexaenoic acid (HDoHE) were lower in the livers of mice with diet-induced NAFLD [55]. In general, oxylipins derived from n-6 PUFAs are considered as pro-inflammatory, while oxylipins derived from n-3 PUFAs are considered to be less-inflammatory or anti-inflammatory [9,19]. Due to the fact that oxylipins mediate diverse biologic activities [9,10,12,13,14], it is conceivable that certain oxylipins might contribute to the pathogenesis of liver toxicity. For example, in our study, AA-derived thromboxane (TX)B_2_ was higher in the livers of chemotherapy-treated animals. TXB_2_ is a stable, inactive, and non-enzymatically driven degradative product of highly unstable TXA_2_ [56,57,58]. In NAFLD, NASH, and other liver diseases, higher serum levels or urinary excretions of TXB_2_ progressively increases with the degree of disease progression [59,60,61,62]. The mechanisms by which enhanced thromboxanes cause liver injury are unknown; however, it is believed that thromboxanes may cause or aggravate liver injury by decreasing blood flow and increasing inflammation and extracellular matrix deposition [63,64,65]. The inhibition of thromboxane production has been shown to alleviate liver injury in several animal models [66,67,68]. Thromboxane inhibition results in the attenuation of necro-inflammatory changes accompanied by a decrease in TNF-α mRNA [63]. Chemotherapy also increases 12-HETE, which is also a well-known pro-inflammatory oxylipin [69,70]. When TXB_2_ + 12-HETE were injected into mice, the productions of TNF-α and IL-6 were significantly elevated in the macrophages isolated from bone marrow [71]. The exact roles of these pro-inflammatory oxylipins in chemotherapy-associated liver toxicities are not known and need to be investigated in future studies.

Seven n-3 PUFA-derived oxylipins are reduced in the liver by chemotherapy treatment. Prior work revealed that n-3 PUFAs are depleted in the liver by chemotherapy treatment [35], although the tissue levels of oxylipins do not always reflect PUFA precursors (our lab and [25]). Oxylipins derived from n-3 PUFAs are thought to modulate inflammation via two pathways: through their direct anti-inflammatory activities, and/or through their proresolving activities [72,73,74]. N-3 PUFAs-derived oxylipins compete for the same receptors as n-6 PUFAs-derived oxylipins, which reduces their concentrations and therefore their inflammatory processes [75,76]. Inflammation is normally terminated by a multistep resolution mechanism [77], which is important for normal homeostasis. This homeostasis mechanism in the tissue is dysregulated in disease conditions [74,78]. Proresolving oxylipins such as resolvins, protectins, and maresins produced from n-3 PUFAs initiate pathways that signal the termination of an acute inflammatory phase [72], and thus play important roles in this resolution phase of the inflammatory process [77]. We found that several hepatic oxylipins such as 14-HDoHE and 17-HDoHE, precursors of resolvins, protectins, and maresins [27,79], were lowest after chemotherapy treatment, which may favor the production of more pro-inflammatory oxylipins. Understanding the role of specific or groups of these oxylipins in inflammation and its resolution may therefore shed light on how n-3 PUFAs mediate their beneficial effects in chemotherapy-associated liver toxicities.

Although the effect of fish oil on oxidants and antioxidants is always controversial, a recent systematic review and meta-analysis of clinical trials showed that n-3 PUFAs-supplementation enhances the antioxidant defense against reactive oxygen species [80]. The provision of dietary fish oil initiated concurrently with chemotherapy treatment significantly reduces oxidative stress markers, as is indicated by the amelioration of hepatic GSSH and 4-HNE that were elevated in the chemotherapy-treated animals (Figure 2A, B). This is important because hepatotoxicity from cytotoxic chemotherapy, at least in part, is related to the increase oxidative stress [81]. Our findings are consistent with findings from other studies, which indicates that n-3 PUFAs act as antioxidants and reduce reactive-oxygen-species-production by eliminating superoxide [82,83], by increasing expression and activity of key antioxidant enzymes [84], or by lowering 4-HNE-modified protein adduct [84]. Thus, improving antioxidant status by fish oil provision can have very positive effects on improving the pathological status in CASSH. This study shows a strong dietary effect on hepatic oxylipin alterations by dietary fish oil. In general, total, as well as individual, n-6 PUFA-derived oxylipins were lower in the livers of animals provided a fish oil diet compared with other animals. Although there is no data available about the direct effect of fish oil on the liver oxylipins in healthy or diseased animals, nor in the livers of chemotherapy-treated animals, the reduction of n-6 PUFA-derived oxylipins in response to dietary EPA or DHA has been observed in the livers of healthy rats [25,85]. As is consistent with our findings, the effects of fish oil on a larger range of plasma n-6 oxylipins have been reported in humans [86,87], as well as the effects of EPA or DHA in animal models [25,84] or in in vitro models [88]. In contrast with the reduction in the n-6-derived oxylipins, total, as well as individual, n-3-derived oxylipins, particularly EPA- and DHA-derived oxylipins, were increased with fish oil feeding. Fish oil provisions resulted in the reduction of ALA oxylipins, suggesting that the retro-conversion of EPA and DHA to ALA did not occur to a significant extent in the livers in this model. Our findings are consistent with the previous study in which dietary fish oil increased kidney EPA- and DHA-derived oxylipins while reducing ALA-derived oxylipins [26]. These dietary effects on oxylipins may have implications for liver injury and recovery. As discussed in previous sections, AA-derived thromboxanes and several HETEs are involved in the inflammatory responses in the liver, while resolvins, protectins, and maresins formed from EPA and DHA have anti-inflammatory effects. Hence, oxylipin alterations in response to diet are likely to impact liver toxicity and repair.

Under normal physiologic conditions, all tissues, including livers, constitutively produce a minimal level of cytokines. When stimulated, liver cells, particularly immune cells in the liver (i.e., Kupffer cells), become activated, and cytokine production increases dramatically to direct the healing process. However, if the production of pro-inflammatory cytokines does not resolve after a short time, damage to healthy tissue occurs and results in toxicity [89,90,91]. In this study, chemotherapy treatment significantly increased hepatic TNF-α, IL-6, and IL-10 levels, while dietary fish oil reduced IL-6 and eotaxin levels. These findings are consistent with other studies in which higher plasma levels of TNF-α and IL-6 were found in patients with NAFLD or NASH compared with the healthy controls [92,93,94]. TNF-α is a pro-inflammatory cytokine characterized by various biological effects with enhanced expression in the liver and positively correlates with liver steatosis, stiffness, and fibrosis, suggesting that it plays important roles in liver toxicity [89]. Human and experimental studies imply that TNF-α plays roles in development of every setting of liver toxicity, including steatosis, necrosis, apoptosis, and fibrosis (reviewed in ref. [95]). TNF-α also regulates the secretion of IL-6 from Kupffer cells [96]. IL-6 is another pro-inflammatory cytokine with deleterious effects in several inflammatory disorders [97]. The role of IL-6 in liver toxicity remains obscure; however, several studies have reported that hepatocyte IL-6 expression correlates positively with plasma IL-6 levels, the degree of hepatic inflammation, and the stage of fibrosis [98,99,100]. This study also found that hepatic IL-10 was elevated after chemotherapy treatment. IL-10 is an anti-inflammatory cytokine that plays important roles in regulating hepatic inflammation, cell necrosis, apoptosis, and liver fibrosis and in stimulating liver regeneration after injury [101,102]. Our findings are consistent with a study that shows serum IL-10 levels were elevated in hepatocellular carcinoma patients compared with normal controls, but that they decreased after surgical resection [103]. The higher hepatic levels of IL-10 with the concomitant increase of TNF-α and IL-6 after chemotherapy treatment might be due to the body’s adaptive mechanisms for the resolution of acute inflammatory responses to protect the liver [104]. Patients with NAFLD exhibit a balance between pro-inflammatory and anti-inflammatory cytokines [89], and these balances are altered with the progression of disease. Whether this higher hepatic level of IL-10 persists after several cycles of chemotherapy remains to be investigated in future studies.

Dietary fish oil mitigates elevated hepatic IL-6. Our finding is consistent with previous findings in which fish oil supplementation significantly reduced serum IL-6 levels in patients with NAFLD [92], as well as patients with other diseases [92,105,106]. The provision of fish oil also reduced hepatic eotaxin levels. Eotaxin is a potent eosinophil-specific chemoattractant which plays an important role in innate and adaptive immune responses [107]. Serum eotaxin levels have been reported to be higher in patients with chronic liver diseases, and this higher serum level was negatively correlated with the liver’s biosynthetic capacity, while being positively associated with higher serum IL-6, hepatic necroinflammation and fibrosis [108]. Similarly, serum eotaxin level has been positively associated with pro-inflammatory cytokines (TNF-α and IL-6) [109,110] and the severity of liver steatosis in NAFLD [111]. Although hepatic eotaxin was not significantly impacted by chemotherapy, the reduction of eotaxin levels by dietary fish oil may favor the liver’s regeneration or resolution mechanisms. However, the exact role of eotaxin in chemotherapy-associated liver toxicities needs to be investigated in future studies.

It should be noted that results obtained from animal models cannot simply be extrapolated to humans due to the fact that lipid metabolism differs between species. However, the experimental model used for this study carefully recapitulated therapy for colorectal cancer in humans with respect to the doses, cycles, and toxicity of a combined regimen of CPT-11+5-FU [37,38,112]. Moreover, we observed a strong dietary effect even after a short time of dietary intervention initiated at the same time as chemotherapy treatment and even after few days of lower food intakes. Female animals were used in this study to enable the assessment of tumor growth and the reduction of variability due to sex. Due to the fact that oxylipin levels and distributions are affected by sex [25,26,40], the results described in the present study may not be (completely) transferable to male animals. Our study is relatively short as only one cycle of chemotherapy was provided, whereas in clinical settings four or more cycles of chemotherapy are commonly provided before liver resection. Therefore, the alteration of oxylipins and the effect of fish oil after several cycles of chemotherapy need to be investigated in future studies. The method we used herein can detect and quantify > 160 non-esterified (free) oxylipins, as the hydrolysis procedures that are currently used to extract esterified oxylipins result in the degradation of certain classes of oxylipins and the potential formation of artifacts, thus measuring the non-esterified oxylipins is the preferred method [113]. There are several methods for oxylipin extraction which vary in their efficiency, with some favoring some types of oxylipins over others; however, the method used herein is one of the most efficient and most frequently used [114]. Several oxylipins have chiral carbons, and consequently present enantiomers that can oppose biological effects [115]. Our method does not provide information on enantiomer balance within an oxylipin species, but rather presents the overall comprehensive oxylipin profile that results from chemotherapy and the modification by a diet containing EPA and DHA. Our data will guide future research to explore targeted oxylipins in these conditions.

## 5. Conclusions

The present study provides a comprehensive hepatic oxylipin profile in normal, tumorous, chemotherapy-treated animals provided a control diet that mimics a western diet, and chemotherapy-treated animals provided fish oil diet. Chemotherapy-induced liver injury is associated with a reduction in oxylipins that are considered anti-inflammatory or less inflammatory with the subsequent increase of pro-inflammatory cytokines. Dietary fish oil mitigated these alterations. Further studies in other experimental models is warranted to verify whether these chemotherapy-induced oxylipin perturbations occur in humans, in both sexes, and whether fish oil supplementation can abrogate those perturbations.

## Figures and Tables

**Figure 1 cancers-14-05703-f001:**
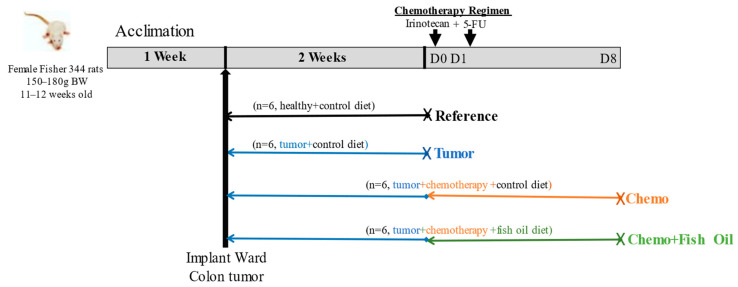
Study design. D, day; FU, fluorouracil; X, kill.

**Figure 2 cancers-14-05703-f002:**
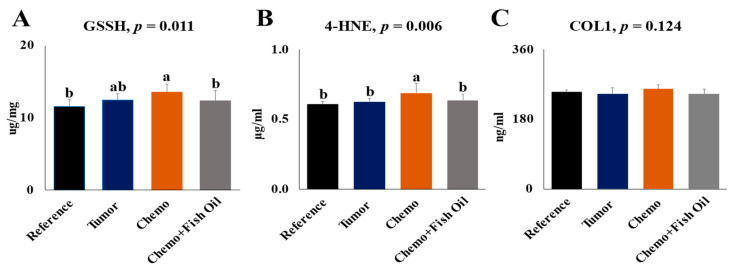
Liver-oxidized glutathione, GSSH (**A**); 4-Hydroxy-2-Nonenal, 4-HNE (**B**); type-I collagen, COL-1 (**C**). Values are expressed as mean ± SD. Significant differences were determined using one-way ANOVA. Post hoc analysis was carried out by Duncan multiple-comparison tests for simple effect when interactions were present or when the Kruskal–Wallis test indicated the presence of differences. Differing lower-case superscript letters indicate significant simple-effect differences between values (a > b). Groups: Reference, healthy rats that did not undergo tumor implantation nor receive chemotherapy, consumed control diet only; Tumor, tumor-bearing animals that did not receive chemotherapy, consumed control diet only; Chemo, tumor-bearing animals that received chemotherapy, consumed control diet only; Chemo+Fish Oil, tumor-bearing animals that received chemotherapy, consumed fish oil diet.

**Figure 3 cancers-14-05703-f003:**
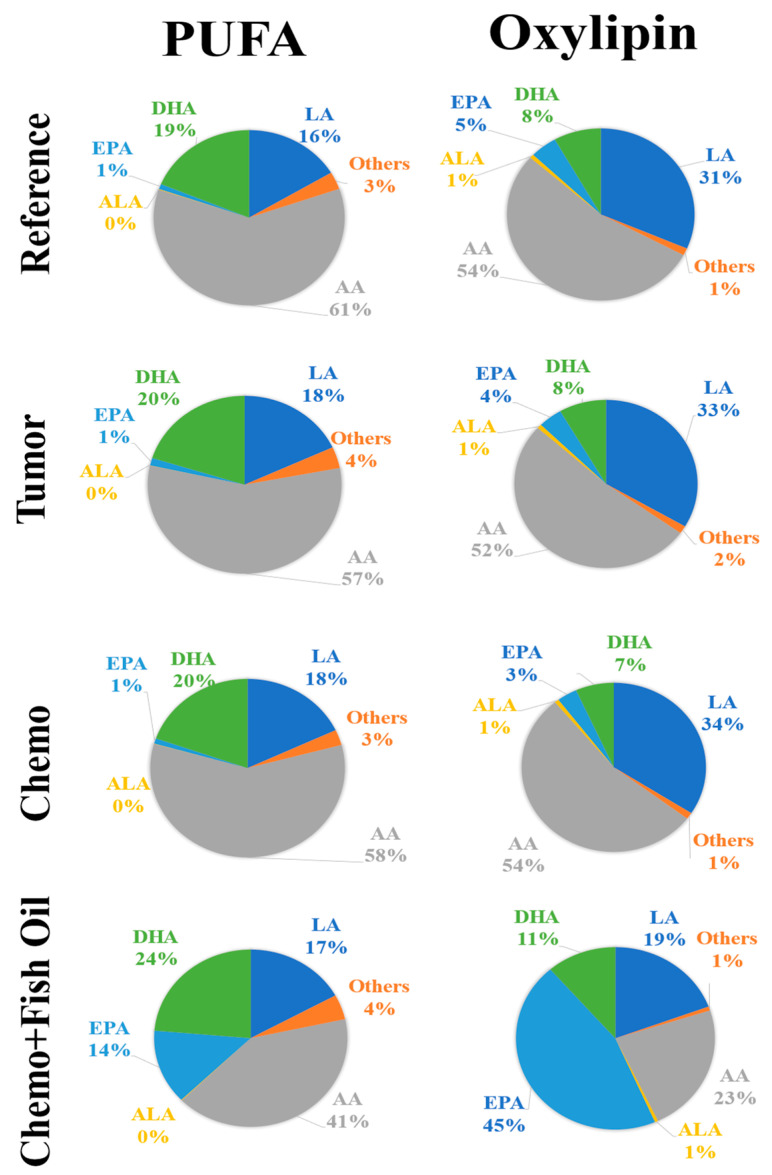
Distributions of phospholipid polyunsaturated fatty acids and oxylipins in the livers of rats provided control and fish oil diets. Groups: Reference, healthy rats that did not undergo tumor implantation nor receive chemotherapy, consumed control diet; Tumor, tumor-bearing animals that did not receive chemotherapy, consumed control diet; Chemo, tumor-bearing animals that received chemotherapy, consumed control diet; Chemo+Fish Oil, tumor-bearing animals that received chemotherapy, consumed fish oil diet, started same day as chemotherapy.

**Figure 4 cancers-14-05703-f004:**
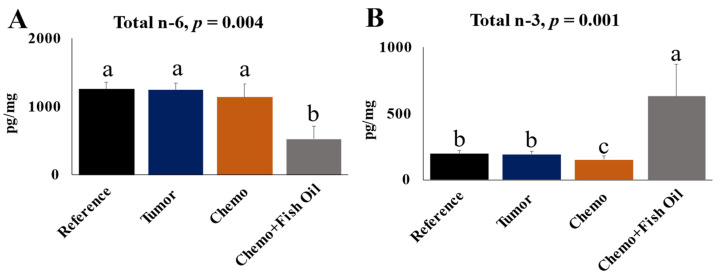
***Liver*** total n-6 oxylipins (**A**); total n-3 oxylipins (**B**). Values are means ± SD, *n* = 6. Significant differences (*p* < 0.05) were determined using one-way ANOVA. Post hoc analysis was carried out by Duncan multiple-comparison tests for simple effect when interactions were present or the Kruskal–Wallis test indicated the presence of differences. Differing lower-case superscript letters indicate significant simple effect differences between values (a > b > c). Groups: Reference, healthy rats that did not undergo tumor implantation nor receive chemotherapy, consumed control diet only; Tumor, tumor-bearing animals that did not receive chemotherapy, consumed control diet only; Chemo, tumor-bearing animals that received chemotherapy, consumed control diet only; Chemo+Fish Oil, tumor-bearing animals that received chemotherapy, consumed fish oil diet.

**Figure 5 cancers-14-05703-f005:**
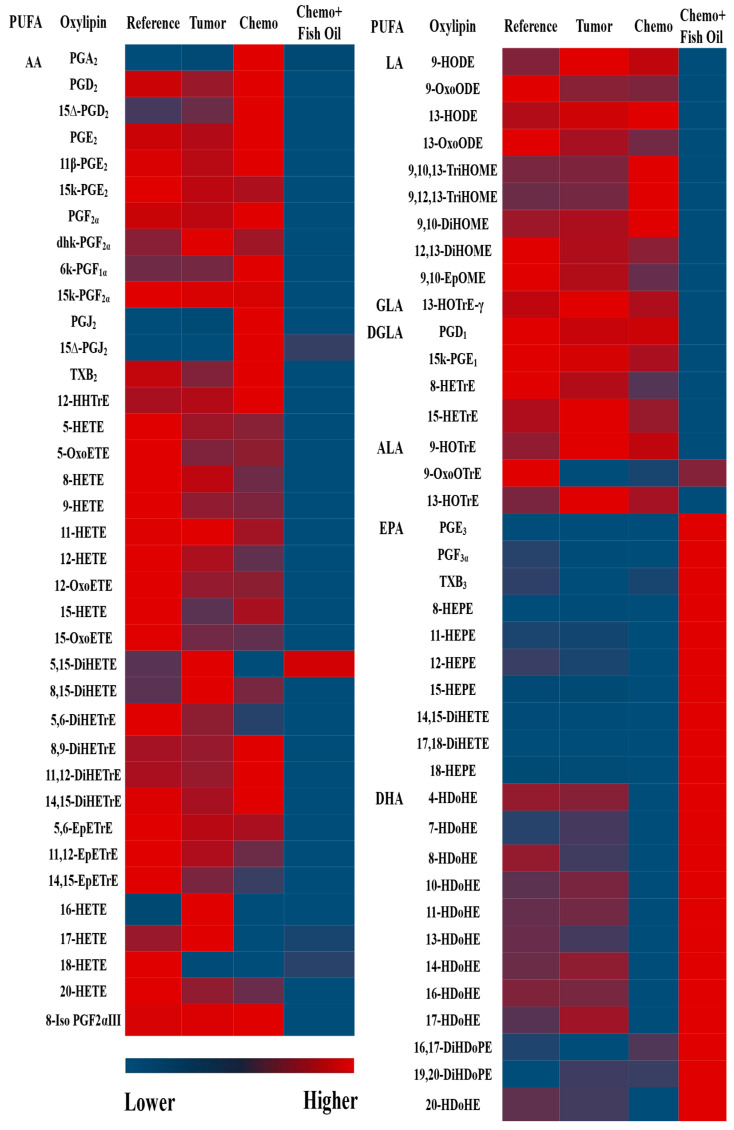
Heatmap of liver oxylipins showing the strong effects of dietary fish oil containing eicosapentaenoic acid (EPA) and docosahexaenoic acid (DHA). Groups: Reference, healthy rats that did not undergo tumor implantation nor receive chemotherapy, consumed control diet; Tumor, tumor-bearing animals that did not receive chemotherapy, consumed control diet; Chemo, tumor-bearing animals that received chemotherapy, consumed control diet; Chemo+Fish Oil, tumor-bearing animals that received chemotherapy, consumed fish oil diet, started same day as chemotherapy.

**Figure 6 cancers-14-05703-f006:**
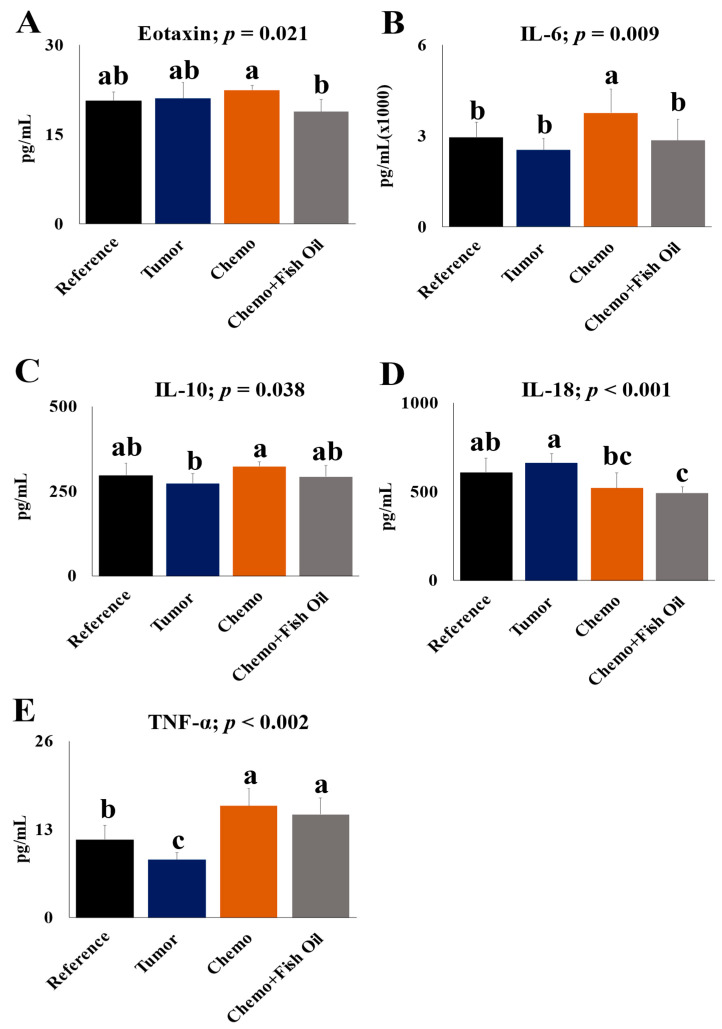
Hepatic eotaxin (**A**); IL-6 (**B**); IL-10 (**C**); IL-18 (**D**) and TNF-α (**E**). Significant differences were determined using one-way ANOVA. Post hoc analysis was carried out by Duncan multiple-comparison tests for simple effect when interactions were present or the Kruskal–Wallis test indicated the presence of differences. Differing lower case superscript letters indicate significant simple effect differences between values (a > b > c). Groups: Reference, healthy rats that did not undergo tumor implantation nor receive chemotherapy, consumed control diet only; Tumor, tumor-bearing animals that did not receive chemotherapy, consumed control diet only; Chemo, tumor-bearing animals that received chemotherapy, consumed control diet only; Chemo+Fish Oil, tumor-bearing animals that received chemotherapy, consumed fish oil diet.

## Data Availability

The data presented in this study are available on request from the corresponding author.

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
