# Peer review of "Dietary EPA+DHA Mitigate Hepatic Toxicity and Modify the Oxylipin Profile in an Animal Model of Colorectal Cancer Treated with Chemotherapy"

_cancers, 2022, doi:10.3390/cancers14225703_

Round 1
Reviewer 1 Report
Summary
These paper aims to describe the effects of fish oil on chemotherapy induced alterations in cytokines and oxylipins. Authors show that fish-oil counteracts the chemotherapy induced alterations and hypothesize that fish oil may attenuate observed alterations in hepatotoxicity. The paper adds to our understanding of chemotherapy induced alterations. Main drawbacks concern the short treatment of rats. It is unclear whether the acute alterations observed are maintained during further chemotherapy cycles. In addition, if results can be easily translated into the clinic and the human situations remains unclear at present. Still this paper adds to our current knowledge of hepatotoxicity.
Introduction: A well written and structured description of the envisioned problem with hypothesis and link to the clinical relevance of this study. References are in general contemporary and up to date.
Suggestion: In what percentage of patients does irinotecan (CPT-11) + 5-fluorouracil (5-FU) result in CASSH? To what extent does it limit de treatment. It would be of interest to have data (if available) on these question to further strengthen the importance of the study
Material and methods: Clearly described procedures. Use of adequate methods to detect markers of interest. There are some minor comments (see below). Statistical paragraph is sufficiently described.
· For me it is not totally clear why the timepoint of 1 week was chosen to stop the study and terminate the animals. Could authors please describe at some point in the manuscript why this cut-off point was chosen?
· The tumor was introduced in the flank (reference 35), was it at macroscopic/microscopic examination also present within the liver (such as would be the case in humans), could this difference affect observed results in the current study?
· Can authors describe how they calculated the adjusted p value cut-off of 0.017 (Bonferroni correction). It seems a quite high cut-off considering the high number of comparisons made within the statistical analysis.
Results: Results are described in detail. Some tables and figures seem superfluous. Figure 5 is a clear overview of the findings, maybe table 1 would be better suited in the supplementary materials.
Discussion: The discussion is well written and main results are clearly displayed. Authors identified the most important pitfalls of their study and describe these pitfalls in relation to their results. The direct translation of their results to the clinic is not straightforward as authors state as well. Nonetheless the presented results show that fish oil may counteract the alterations induced by chemotherapy and may thus contribute to our understanding of chemotherapy induced alterations. Analysis of these markers after more cycles of chemotherapy and development of liver toxicities in these animals is out of the scope of this paper but could generate stronger evidence for authors hypothesis.
Abstract: Clear description of study and main results.
Figures:
Table 1: Please describe which groups are a, b and c.
Figure 1: lowering the position of the healthy control diet rats below the time bar would maybe make it even more clearly that there were 24 animals within this study.
Figure 2: In general, the figure is clear. The additional Duncan multiple comparisons’ test to identify differences among specific groups denoted with a and b seem a bit unclear. Please make these observed differences clearer either in the figure or replace by a table.
Figure 5 and Table 1 seem to have quite the same data: Authors could consider to move either the table or the figure to the supplementary materials.
Minor comments:
Line 124: Please change “suppled with the kit” to “supplied with the kit”
Paragraph 2.2: line 122-130: please state if all measurements were done according to manufactures instructions or if alterations were made to the protocol.
Contributions and funding and ethics: Appropriately described.
Reviewer 2 Report
The manuscript entitled “Dietary EPA+DHA Mitigate Hepatic Toxicity and Modifies Oxylipin Profile in an Animal Model of Colorectal Cancer Treated with Chemotherapy” submitted to Cancers by Dr. Monirujjaman and co-workers analyses the role of EPA and DHA on lipid and oxylipin profile and cytokine release in an experimental model of chemotherapy-associated steatosis/steatohepatitis. The study presents a lot of interesting findings but I offer some comments/suggestions that can improve the manuscript.
Line 63. “Depending on the PUFA substrate, oxylipins can be classified as pro- or anti inflammatory [8, 17-19]” Please checking this sentence. I believe that the pro-inflammatory or anti-inflammatory effect of oxylipins depend on its biological effects.
Line 67. “Dietary supplementation with specific n-3 PUFAs (i.e., eicosapentaenoic acid, EPA and docosahexaenoic acid, DHA) increases cellular membrane content of these fatty acids [23-25]”.
Line 146. “A list of all oxylipins screened for, detector response factors and internal standards are listed in [38-40]. Detection and quantification limits were set at 3 and 5 levels above background, respectively. Quantification of oxylipins was determined using the stable isotope dilution method [42], and amounts expressed as pg/mg of tissue”. These references present this important information in supplementary tables (“Further details of oxylipins scanned for but below the limit of detection (b3 times above baseline) or below the limit of quantitation (b5 times above baseline), as well as retention times, mass transitions, internal standards and standard curve slopes are provided in Supplemental Tables S1 and S2) but unfortunately, I was not able to meet part of this information in these tables. After to study the retention times (really very near) I have some doubts on the possibility of a good quantification. I believe that the presentation of this methodologies should be improved (see doi: 10.1016/j.jpba.2011.06.01; doi: 10.1016/j.jchromb.2014.05.024.).
In the above-mentioned references (38-40) oxylipins were expressed as pg/mg of tissue but here were expressed as ng/g tissue.
The findings show in table 1 and figure 5 appear redundant.
Figure 6A. Why Mean (pg/ml)? I believe that Eotaxin (pg/ml) is more appropriate. The same along the figure.
Line 318. “Cytotoxic chemotherapy” or Chemotherapy?
Line 377. “Our findings are consistent with findings from other studies which indicates that n-3 PUFAs may act indirectly as an antioxidant, reduce reactive oxygen species (ROS) production by eliminating superoxide [79, 80] or by increasing expression and activity of key antioxidant enzymes [81]. Similarly, fish oil supplementation lowers 4-HNE-modified protein adduct concentration in the human atrial tissue”. This paragraph is incongruity. The impairment of ROS production is not an indirect antioxidant effect in my opinion. There are bibliography on the pro-oxidative effects of EPA-DHA/fish oil that should be considered in the discussion.
Several oxylipins have chiral carbons and consequently present enantiomers that can have oppositive biological effects. Cabral and co-workers reported that R HODEs are mitogenic whereas S HODEs are apoptotic (doi: 10.1152/ajpgi.00064.2014). Here, we observed dramatically changes of oxylipins levels without information on R/S enantiomer balance. In the context of the study, this aspect can be important to explore.
Round 2
Reviewer 2 Report
In my opinion some aspects of the oxylipin analysis are unclear described yet. Some methodology aspects are presented in supplementary material of a reference of a reference.
EPA supplementation (fish oil) increased EPA content in PL and consequently increased oxylipins from EPA (Figure 3) induced a markedly decrease of oxylipins from AA. These findings have been previously reported (ex. doi: 10.1002/jcp.20678) and this important aspect should be discussed in a more profound way.
